# A Single-Center, Randomized Controlled Trial to Test the Efficacy of Nurse-Led Motivational Interviewing for Enhancing Self-Care in Adults with Heart Failure

**DOI:** 10.3390/healthcare11050773

**Published:** 2023-03-06

**Authors:** Federica Dellafiore, Greta Ghizzardi, Ercole Vellone, Arianna Magon, Gianluca Conte, Irene Baroni, Giada De Angeli, Ida Vangone, Sara Russo, Alessandro Stievano, Cristina Arrigoni, Rosario Caruso

**Affiliations:** 1Department of Public Health, Experimental and Forensic Medicine, Section of Hygiene, University of Pavia, 27100 Pavia, Italy; 2Department of Biomedicine and Prevention, University of Rome Tor Vergata, 00133 Rome, Italy; 3Department of Nursing and Obstetrics, Wroclaw Medical University, 50-368 Wrocław, Poland; 4Health Professions Research and Development Unit, IRCCS Policlinico San Donato, 20097 San Donato Milanese, Italy; 5Nursing Degree Course, Section Istituti Clinici di Pavia e Vigevano S.p.a., University of Pavia, 27100 Pavia, Italy; 6Department of Clinical and Experimental Medicine, University of Messina, 98122 Messina, Italy; 7Centre of Excellence for Nursing Scholarship, OPI, 00136 Rome, Italy; 8Department of Biomedical Sciences for Health, University of Milan, 20133 Milan, Italy

**Keywords:** education, heart failure, motivational interviewing, randomized clinical trial, self-care

## Abstract

Background: The role of nurse-led motivational interviewing (MI) in improving self-care among patients with heart failure (HF) is promising, even if it still requires further empirical evidence to determine its efficacy. For this reason, this study tested its efficacy in enhancing self-care maintenance (primary endpoint), self-care management, and self-care confidence after three months from enrollment in adults with HF compared to usual care, and assessed changes in self-care over follow-up times (3, 6, 9, and 12 months). Methods: A single-center, randomized, controlled, parallel-group, superiority study with two experimental arms and a control group was performed. Allocation was in a 1:1:1 ratio between intervention groups and control. Results: MI was effective in improving self-care maintenance after three months when it was performed only for patients (arm 1) and for the patients–caregivers dyad (arm 2) (respectively, Cohen’s d = 0.92, *p*-value < 0.001; Cohen’s d = 0.68, *p*-value < 0.001). These effects were stable over the one-year follow-up. No effects were observed concerning self-care management, while MI moderately influenced self-care confidence. Conclusions: This study supported the adoption of nurse-led MI in the clinical management of adults with HF.

## 1. Introduction

Heart failure (HF) is a major public health concern worldwide, affecting approximately 1–2% of the global adult population [1]. HF is a clinical syndrome caused by several potential underlying etiologies and characterized by key symptoms such as dyspnea, ankle swelling, exhaustion, and clinical signs (e.g., peripheral edema) [2]. HF is associated with poorer quality of life, increased hospitalization rates, more health-related costs, and decreased overall survival in patients [1,3,4]. It also places health-related challenges on the well-being of informal caregivers, because it is associated with a reduced quality of life and health-related issues [5].

Patients with HF need to adhere to the recommended medication regimen and pay special attention to dietary sodium and liquids restrictions, exercise regimen, body condition monitoring, behaviors and mood control, accurate symptom detection, therapy impact evaluation, and other self-care behaviors [2,6]. These demands are often mismatched from the required self-care practices, as the self-care behaviors of adults with HF were extensively described as mainly inadequate [7,8,9,10,11,12].

Self-care is the decision-making process that includes behaviors that help maintain heart failure stability (self-care maintenance), allow patients to perceive symptoms (self-care monitoring), and manage signs and symptoms (self-care management) [13,14]. Self-care maintenance includes exercising (e.g., brisk walking), avoiding getting sick, medical adherence, and dietary and liquids adherence. Self-care monitoring is based on promptly recognizing the cardinal HF symptoms and signs (e.g., gaining weight, dyspnea, peripheral edema). Self-care management reflects patients’ knowledge and health literacy in decision-making when symptoms and/or signs occur. Overall, self-care behaviors are positively influenced by the patient’s perception of adequately performing demanding self-care behaviors (self-care confidence) [13,14]. 

Among the strategies to sustain adequate self-care in patients with HF, motivational interviewing (MI) showed promising results [15,16,17,18,19]. By exploring and resolving ambivalence, MI, a goal-directed and patient-centered counseling technique, assists individuals in improving their health-related behaviors [16,20,21,22]. The essential components of MI include showing empathy, creating discrepancies in the perceptions derived from interpreting the gap between expected behaviors and unhealthy performed ones, refraining from disagreements, promoting self-efficacy, and sustaining a shared strategy [23]. Individual psychosocial behavioral interventions utilizing MI showed improved medication adherence and high levels of participant satisfaction in several chronic conditions [16,24]. Recent studies show that nurse-led MI is safe and effective in pursuing behavioral changes in patients with chronic conditions because nurses are healthcare professionals who work closely with the patient’s needs, beliefs, and behaviors, and are able to detect misconceptions regarding clinical aspects [24,25].

A recent meta-analysis of nine experimental studies shows that MI has moderate effects on enhancing self-care confidence and self-care management and large effects on improving self-care maintenance [26]. Despite this important synthesis of evidence, the authors stated that more empirical and experimental research is still required to corroborate the efficacy of MI on self-care in patients with HF because of the current heterogeneity in the several included populations and the poor adoption of clinical trials measuring self-care with theory-grounded self-report scales [26]. In other words, more randomized controlled trials are required to close the gap of evidence that currently undermines the generalizability and transferability of the efficacy of MI in managing HF [27]. For this reason, this randomized clinical trial (RCT) aimed (a) to test the efficacy of nurse-led MI in enhancing self-care maintenance (primary endpoint), self-care management, and self-care confidence after three months from enrollment in adults with HF compared to usual care, and (b) assess changes in self-care over follow-up times (3, 6, 9, and 12 months). 

## 2. Materials and Methods

### 2.1. Design

This was a single-center, randomized, controlled, parallel-group, superiority study with two experimental arms and a control group. Allocation was based on a 1:1:1 ratio between intervention groups and control. The ClinicalTrial.gov identifier is NCT05595655. This study was approved by the Ethical Committee of San Raffaele Hospital (approval #74/INT). 

### 2.2. Study Setting

This study enrolled ambulatory patients in the Heart Failure Clinic of the IRCCS Policlinico San Donato in northern Italy. The focus of care ranges from prenatal diagnosis to rehabilitation, from newborns to the very elderly; the medical–nursing staff is specialized in several areas of cardiology, heart surgery, vascular surgery, and anesthesia with a high focus on clinical research [28]. IRCCS Policlinico San Donato is a reference center for cardiovascular diseases [29].

### 2.3. Participants

Participants were patients with HF who did not practice adequate self-care and their caregivers. Patients met the requirements for participation if they met the following criteria: (a) had a diagnosis of HF classified as New York Heart Association (NYHA) class II–IV; (b) had evidence of inadequate self-care determined by a score of 0, 1, or 2 on at least two items of the self-care maintenance or self-care management scales of the Self-Care of HF Index v.6.2 (SCHFI v.6.2) [30]; (c) were willing to sign the informed consent to be enrolled; and (d) with age ≥ 18 years. Patients who had a myocardial infarction during the previous three months and/or had severe cognitive impairment with a six-item Screener score between 0 and 4 [31] and/or residing in a nursing home where self-care was not required or had an informal caregiver who did not wish to be involved in the study were all excluded from the study. Informal caregivers were eligible to be enrolled if the patients confirmed them as the principal caregivers. Both were not eligible to be enrolled if either the patient or the caregiver refused to participate in the trial in the baseline period; however, if one participant left the study after enrollment, the other one was allowed to continue. Eligible dyads were enrolled after having received a clinician invitation letter stating the aim of the study and the procedure. 

### 2.4. Experimental Arms

A trained nurse with experience in educating patients with HF delivered MI. Four registered nurses were trained to participate in a 32 h training course on MI and 8 h refresh training on evidence-based care regarding HF. The registered nurses were females; two of them had a Master of Science in nursing, one was a doctoral student (PhD student) in nursing science, and one had a bachelor’s degree. The nurses’ average age was 28.75 years (standard deviation, SD = 5.12; range: 24–36). They had 5.75 years of work experience in cardiology (SD = 4.35; ranges, 2–12). The intervention included face-to-face nurse-led MI interventions that lasted around 30 min. The first MI had to be performed within 2 months from enrolment and followed by four other MI interventions at 3, 6, 9, and 12 months performed by the same interventionist. To strengthen the intervention and to sustain adherence to the protocol, the nurse who performed the MI contacted the patients via telephone three times during the first two months after MI. This scheduled approach for delivering MI 5 times during the study has never been tested in previous studies [16].

In arm 1, the MI was delivered only to patients; in arm 2, MI was delivered simultaneously to the dyad patient and caregiver. Participants enrolled in the experimental arms (arms 1 and 2) received MI interventions as an add-on approach to the standard of care. 

### 2.5. Standard of Care and Control Group

Standard of care included clinical visits in the outpatient settings every 6 to 12 months, depending on the severity of the patients’ HF conditions and their specific clinical pathways. Education in the standard of care was based on discussions with patients about relevant materials geared toward HF self-care. Patients in the control group received standard of care only.

### 2.6. Procedures

A research assistant (outcome assessor) screened the patients using the SCHFI v.6.2 [30] and the six-item Screener [31] following the study protocol after patients and caregivers gave their consent. After the eligibility screening, when a patient was eligible, the protocol-required questionnaires were administrated to both patients and caregivers. They received questionnaires individually at baseline and at each follow-up, and they were not permitted to work together to complete the questionnaires. At 3, 6, 9, and 12 months after enrollment, follow-up data were collected via telephone. The outcome assessor was kept blind regarding the research arms at both the baseline and all follow-up points. Interventionists and participants were not blind to the study arm.

### 2.7. Randomization

A web-based system generated the randomization sequence, assigning participants in a 1:1:1 ratio to either the intervention or control group using a simple randomized process. Allocation sequences were accomplished using computer-generated algorithms that were made available after the trial. The interventionists were not informed of the allocation sequence. The randomization process started after the site employees (study nurses) entered the patients’ information into the database (RedCap). Each randomization number was generated and sent by a study nurse to the interventionist (a trained nurse who performed the MI), who was not the professional who had to assess the outcomes. Each participant’s enrollment and follow-ups were always communicated to the trial coordinator.

### 2.8. Measurements

The measurements for patients were socio-demographic and clinical characteristics. Socio-demographics were sex (male, female); age (years); marital status (single, married, divorced, widower); education (high schools or higher, lower than high schools); employment (active worker, retired); income (more than necessary to live, the necessary to live, and not the necessary to live). Clinical characteristics were NYHA class (II, II, IV functional class), Charlson comorbidity index (CCI, score) [32], ejection fraction (HFpEF = preserved ejection fraction; HFmrEF = midrange ejection fraction; HFrEF = reduced ejection fraction), time with HF (months), BMI (kg/m^2^), Montreal cognitive assessment (MoCA) (score) [33]. The outcomes of this study were the self-care maintenance scores measured using SCHFI v.6.2 [30]. 

#### Outcomes

The SCHFI v.6.2 was used to assess the score of self-care maintenance at baseline, after 3 months (primary endpoint), and over the follow-up times. The SCHFI v.6.2 also allowed researchers to measure secondary outcomes: self-care management and self-care confidence at baseline and over the follow-up times. Each score has a range of 0 to 100. Higher scores indicate better self-care. Only if a patient had previously reported experiencing HF symptoms, such as dyspnea, did they have to fill out the self-care management scale. A score of less than 70 on each domain denoted adequate self-care.

### 2.9. Sample Size

The pooled mean of self-care maintenance described using the SCHFI v.6.2 in two previous descriptive studies performed in northern Italy was 53.55, with a pooled standard deviation of 18.98 [34]. Previous studies showed that MI could improve the mean of self-care scores in patients with HF by increasing the mean scores with a delta (Δ) ranging from 6 to 15 (pooled mean Δ = 10.95) [26]. Therefore, 49 patients per arm were required to reject the two-tailed null hypothesis of equal mean scores between the study arms with a power of 80%. A sensitivity analysis considering slights variations in the Δ and accounting for 20–25% of attrition as per similar research [16] showed that a total of 180 ± 6 participants was necessary to preserve enough power (80%) to detect significant mean differences between the experimental arms and the control group (60 ± 2 participants per arm).

### 2.10. Treatment Fidelity

The trial coordinator evaluated treatment fidelity by randomly applying an evaluation of the performed MI in arms 1 and 2 using the Motivational Interviewing Treatment Integrity (MITI) Scale [35]. The MI interventions were all audio-recorded, and the MITI was used to randomly evaluate 4 MI interventions per arm at each time point. The scores ranged between 2 and 5, and the median of the assessments in both arms was 3, indicating an ideal technical quality score. 

### 2.11. Timeline

Enrollment required approximately 36 months (from May 2017 to May 2020; the study ended with the last follow-up in May 2021) to avoid overwhelming the activities of the involved staff in the study (i.e., four interventionists, a trial coordinator, two outcome assessors, a study nurse, a data manager, the principal investigator, and the co-investigators). The study was conducted at a cardiovascular hub center that remained operational during the COVID-19 pandemic waves. As a result, the researchers were able to conclude the study during the pandemic by leveraging the center’s ongoing interactions with heart failure patients. Figure 1 shows the patient flow.

### 2.12. Statistical Analysis

All data were analyzed by using an intention-to-treat approach. Categorical variables were described in terms of absolute and relative frequencies. Interval and continuous variables were evaluated for normality using the Shapiro–Wilk test, and data with a normal distribution are presented using the mean and standard deviation (SD). The median and interquartile ranges (IQR) were used to summarize non-normally distributed data. Baseline characteristics were compared between arms to determine if they were equal. Missing scores in the outcomes were 12%, 10%, and 11% (respectively, the extent of the missingness in arms 1, 2, and 3) in each arm, which were imputed by employing multiple imputations based on random effects models after having assessed that the missing mechanisms (missing in relation to time and study arm) and patterns (monotone missingness based on sensitivity analysis) supported the missing at random (MAR) assumptions. 

The delta (Δ) of the self-care scores was calculated at each follow-up period by subtracting the baseline self-care score (T0) to determine the changes in self-care scores during follow-up times (T1, T2, T3, and T4). As the primary endpoint was a significant improvement in arms 1 and 2 of self-care maintenance scores over the control group, a two-sample *t*-test was employed to compare the delta of self-care score in arms 1 and 2 versus the control arm 3, under the assumptions of the central limit theorem [36]. A similar approach was performed for each follow-up time and the secondary outcomes (self-care management and self-care confidence). Precisely, the t-test effect size estimates were computed using d statistics for independent *t*-tests (Cohen’s d), where d values lower than 0.5 indicated small effects, between 0.5 and 0.8 moderate effects, and greater than 0.8 large effects [37]. In addition to this approach, data on the primary and secondary outcomes at each follow-up time were summarized in adequate (scores equal to or greater than 70) or inadequate (scores lower than 70) and compared (arm 1 vs. arm 3; arm 2 vs. arm 3) using chi-square test or Fisher’s exact test when appropriate. 

Mixed models for repeated measures were used to analyze changes across time (from baseline to T4) in primary and secondary outcomes, following the strategy of a previous study [16]. As a dependent variable, these models included the outcome scores available from T0 to T4 for each patient in the study arm. By having included a random intercept in the models, the inter-dependence between self-care maintenance, management, and confidence on the same subject was addressed. The randomization arm (nominal variable) was included in the models as an independent variable, along with the baseline characteristics (i.e., age, sex, income, NYHA, CCI score, MoCA, time since diagnosis, ejection fraction, and self-care confidence). Furthermore, the slopes derived from the models were compared between arms 1 and 2 versus the slopes of arm 3 for each outcome. 

The significance level was set at 0.05 in all tests, and analytics were performed using Stata Statistical Software: Release 17 (StataCorp. 2021; College Station, TX, USA: StataCorp LLC). 

## 3. Results

### 3.1. Participants’ Characteristics

Patients’ baseline characteristics, stratified and compared by arm, are shown in Table 1. No differences are found in relation to the baseline characteristics. The majority of patients were females (r in arms 1, 2, and 3: 51.1%, 55.0%, and 52.5%, respectively) as well as the majority of caregivers (in arms 1, 2, and 3: 73.4%, 69.5%, and 74.0%, respectively). In arms 1, 2, and 3, patients reported mean ages of 68.39 (SD = 12.14), 69.44 (SD = 6.71), and 71.08 (SD = 12.95), respectively. In arms 1, 2, and 3, caregivers reported mean ages of 56.28 (SD = 9.12), 59.44 (SD = 11.10), and 58.17 (SD = 9.08), respectively. In arms 1, 2, and 3, most of caregivers were married: 57.1%, 63.2%, and 61.50%, respectively; for patients, 54.1%, 45.0%, and 37.7%, respectively. Most patients and caregivers reported an educational status lower than high schools: for patients in arms 1, 2, and 3, 72.1%, 75.0%, and 73.8%, respectively; for caregivers, 59.1%, 63.7%, and 62.9%, respectively. 

Specifically, regarding patients, most of them answered that they have the necessary income to live. The median (IQR) time with HF was approximately 4 years in the three arms. The median (IQR) BMI indicated values within normal scores. Overall, most patients were in NYHA II class, with two comorbidities, an HF with preserved ejection fraction (HFprEF) and inadequate self-care maintenance and management scores.

### 3.2. Self-Care Maintenance (Primary Endpoint), Management, and Confidence at the First Follow-Up (T1, 3 Months)

The increase in the self-care maintenance scores (primary endpoint) from baseline to T1 (3 months after enrolment) is higher in arms 1 and 2 compared to arm 3 (Figure 2).

In arms 1, 2, and 3, the mean Δ indicating an increase in the self-care maintenance score is 12.84 (SD = 11.50), 10.81 (SD = 13.05), and 2.78 (SD = 10.33), respectively, indicating a large effect size in the Δ between arm 1 and arm 3 (Cohen’s d = 0.92, *p*-value < 0.001), and moderate effect size in the Δ between arm 2 and arm 3 (Cohen’s d = 068, *p*-value < 0.001). 

Regarding self-care management scores, no differences are found between arm 1 and arm 2 versus arm 3 (see Table 2). Conversely, regarding self-care confidence scores, only the increased scores observed in arm 2 are significantly higher than those in arm 3, with a moderate effect size (Cohen’s d = 058, *p*-value = 0.002). 

The comparisons of the dichotomized scores into adequate (scores ≥ 70) and inadequate (scores < 70) do not show significant differences for each outcome (see Table 2).

### 3.3. Changes in Self-Care Maintenance, Management, and Confidence over Follow-Up Times

The description of self-care maintenance, self-care management, and self-care confidence scores over time are reported in Figure 2 and Table 2. In relation to self-care maintenance, we generally find stability since 1 year (T4) of the effects detected at T1 (after 3 months). No differences are found in relation to self-care management scores over time. Conversely, regarding self-care confidence, at T1 and T2, moderate–small improvements are observed in arm 2 compared to arm 3; in arm 1, self-care confidence shows small–moderate improvements at T3 and T4 (see Table 2).

The trends over time (from baseline to T4) derived from the mixed models in self-care maintenance, self-care management, and self-care confidence scale scores are shown in Figure 3. Regarding the self-care maintenance slopes, arm 1 and arm 2 versus arm 3 show significant differences (*p*-values = 0.038; *p*-values = 0.047, respectively). No differences are found concerning self-care management scores (*p*-values = 0.398; *p*-values = 0.447, respectively). Regarding self-care confidence, only the comparison between trends of arm 2 and 3 show significant differences (*p*-values = 0.031). These trends are confirmed when the mixed models are adjusted for age, sex, income, NYHA, CCI score, MoCA, time since diagnosis, ejection fraction, and baseline self-care confidence.

## 4. Discussion

This study demonstrated that nurse-led MI performed using a scheduled approach (every three months over one year) was effective in improving self-care maintenance with stable effects over the follow-up times. The scheduled approach used to deliver MI in this study is a significant innovation, since no previous randomized controlled trials have utilized a similar approach [6,16,26,38,39,40,41]. This approach allows for a more structured and consistent delivery of motivational interviewing to participants, which may enhance its effectiveness. In this study, nurse-led MI also improved self-care confidence, with some differences when the intervention was performed only for patients (arm 1) or for the dyads of patients and caregivers (arm 2). 

Overall, the results derived from this RCT corroborate previous evidence [6,16,26,38,39,40,41], adding additional insights regarding five main aspects: (a) the nurse-led MI performed with scheduled recurrences over time likely produces stable effects in improving self-care maintenance over time; (b) the characteristics of HF (e.g., NYHA class or ejection fraction) seem to play a non-significant role on influencing the efficacy of MI in improving self-care maintenance over time; (c) the effects of MI performed only for patients seemed to be more stable over the effects showed by performing MI to the dyads in a different way from the effects shown in a previous study [16]; (d) the role of MI in improving self-care management remains unclear; (e) self-care confidence seems positively influenced by MI. 

The efficacy of nurse-led MI on self-care maintenance has important clinical implications because it means that aspects such as treatment adherence, which are highly problematic among patients with HF, might be susceptible to significant improvements when trained nurses employ MI in clinical practice. It is not surprising to find that the nurse-led MI effectively leads patients toward behavioral change [42,43,44]. In this regard, the key features of MI, such as adopting open-ended questions, affirmation of patients’ strengths, adopting reflective listening, and summarizing key points of the discussion, have the potential to be effective in patients with several clinical conditions, from individuals with HFprEF to patients with HFrEF. 

The more stable effects on improving self-care maintenance shown in arm 1 over arm 2 may be explained by the nature of the training performed by the interventionists, which was mainly focused on the elements of MI per se and a brief refresh about evidence-based care for patients with HF rather than focusing on providing the skills to manage the complexity of the dyadic relationships during the MI. In other words, it is reasonable that interventionists found it easier to perform the MI only for patients rather than simultaneously managing the dyad as required in arm 2. In this regard, we have to acknowledge that a previous multicentric RCT found that the effects of MI performed for the patient–caregiver dyad were larger than the MI performed only for patients [16]. From a theoretical perspective, if we consider the contribution of caregivers to the self-care practices of patients with HF [45,46,47], the MI performed for the dyad should be the best option. However, the evidence from this study points out that delivering MI to the dyad should be based on different training from the one designed only to provide skills for delivering MI-based interventions because the complexity of the dyadic relationship should be included in educating the interventionists. 

Among self-care behaviors, self-care management practices seem to be less susceptible to changes than self-care maintenance. This aspect is theoretically explainable by the role of several aspects that determine self-care management, such as disease-specific knowledge and, broadly speaking, health literacy [48,49,50]. In fact, self-care management reflects different individual-level characteristics into actions, from values, beliefs, knowledge, and so on, to behaviors that reflect a decision-making process triggered by the detection of signs and/or symptoms [51,52,53]. Considering these aspects, it is reasonable to think that self-care management requires complex and multiple interventions to be modified (e.g., psychosocial interventions combined with knowledge-based education and MI). Therefore, complex and multiple interventions should aim to affect the main determinants of self-care management rather than self-care management per se. 

Self-care confidence is also susceptible to improvement after MI interventions. Considering that self-care confidence is one of the strongest predictors of self-care behaviors [51,54,55], this result might have interesting clinical implications because improving self-care confidence may trigger virtuous circles to improve several other health-related outcomes. The differences emerging between the two experimental arms of this study (i.e., arm 1 shows effects after six months, while arm 2 shows brief-term effects) require more investigations with future studies and might reflect the complexity of managing MI in a dyadic setting. 

This study has several limitations. First, the single-center design limits the generalizability of the results. Second, the self-report scale used to assess primary and secondary outcomes (SCHFI v6.2) was the best option when the protocol of this RCT was written; however, it is currently outdated because the new SCHFI v7.2 is psychometrically more robust and allows researchers to assess self-care monitoring. Third, patient attrition over the trial was considerably large (19.3% at T4); this aspect requires further mitigation strategies in future studies and a more robust approach to ensure patient adherence to the protocol. Four, the poor focus of the educational course for educating the interventionists regarding managing the complexity of the dyadic relationships between patients and caregivers might be considered a source of bias, especially in interpreting the effects of arm 2. Finally, it is important to interpret the stability of the effects observed in relation to self-care maintenance with caution, given the repeated MI in the experimental procedure. While the results of this study suggest that the repeated MI approach may produce stable effects over a one-year follow-up period, it is important to consider that individual patients may respond differently to repeated interventions. Therefore, the generalizability of the findings to all patients with heart failure should be approached with caution.

## 5. Conclusions

Nurse-led MI shows efficacy in improving self-care maintenance in patients with HF over a one-year follow-up. This RCT confirms previous evidence and supports the adoption of nurse-led MI in the clinical management of HF. Future research should corroborate this evidence in specific subgroups to enhance the external validity of this intervention and should explore the effects of nurse-led MI on clinical outcomes. 

## Figures and Tables

**Figure 1 healthcare-11-00773-f001:**
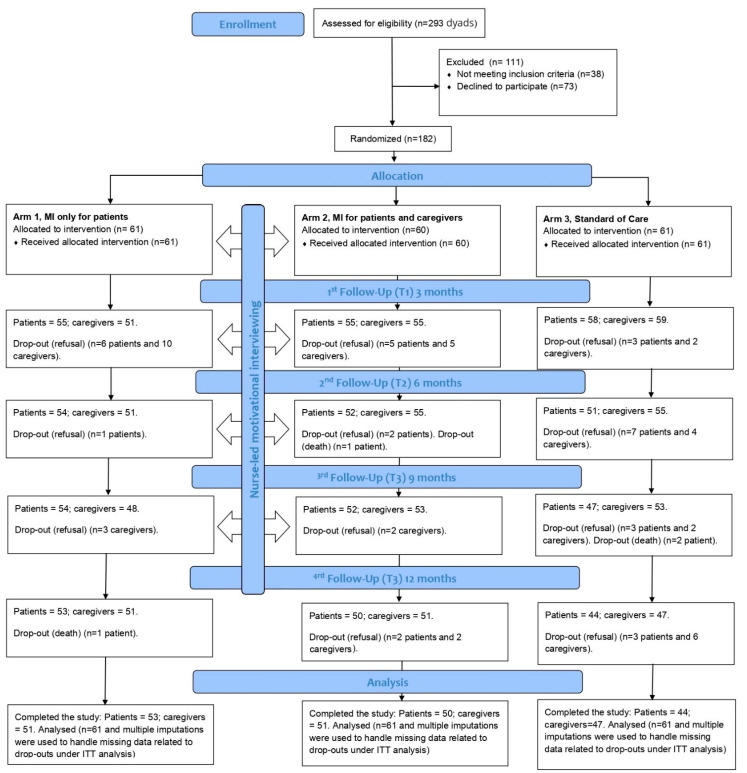
Patient flow.

**Figure 2 healthcare-11-00773-f002:**
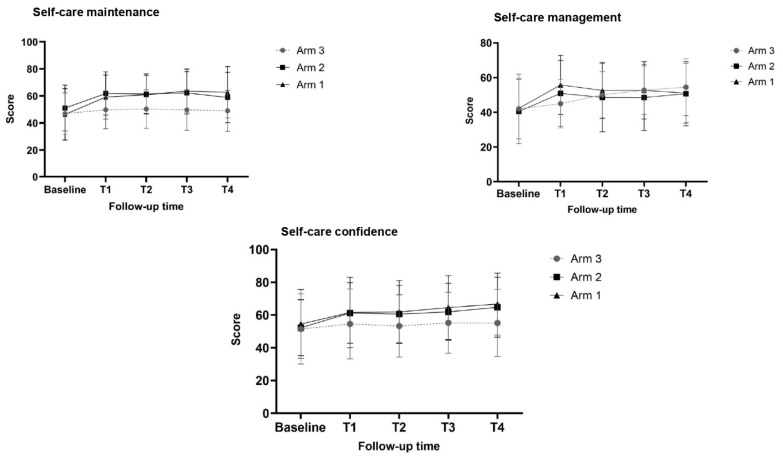
Self-care scores by arms over follow-up times.

**Figure 3 healthcare-11-00773-f003:**
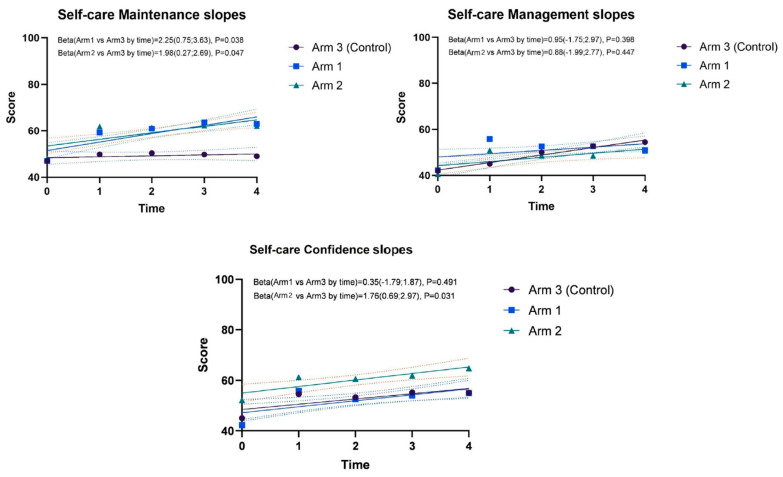
Model-based self-care maintenance, self-care management, and self-care confidence scores by follow-up times.

**Table 1 healthcare-11-00773-t001:** Patients’ characteristics at baseline (n = 182).

		Arm 1: MI Only for Patients (n = 61)		Arm 2: MI for Patients and Caregivers (n = 60)		Arm 3: Usual Care for Patients and Caregivers (n = 61)		*p*-Value
		n	%	n	%	n	%	
Variables								
Sex								
	Males	33	54.1	33	55.0	32	52.5	0.960
Age								
	Years (mean; SD)	68.39	12.14	69.44	6.71	71.08	12.95	0.403
Marital status								
	Single	8	13.1	10	16.7	9	14.8	0.374
	Married	33	54.1	27	45.0	23	37.7	
	Divorced	8	13.1	10	16.7	7	11.5	
	Widower	12	19.7	13	21.7	22	36.1	
Education								
	High schools or higher	17	27.9	15	25.0	16	26.2	0.937
Employment								
	Retired	50	82.0	48	80.0	45	73.8	0.423
Income								
	More than necessary to live	6	9.8	8	13.3	10	16.4	0.264
	The necessary to live	46	75.4	44	73.4	49	80.3	
	Not necessary to live	9	14.8	8	13.3	2	3.3	
Time with HF								
	Months (median; IQR)	48.00	24.00–70.00	47.50	26.75–60.00	46.50	27.00–67.00	0.315
BMI								
	Kg/m^2^ (median; IQR)	24.36	22.00–27.92	25.00	22.75–26.5	25.71	22.00–28.31	0.585
MoCA								
	Score (median; IQR)	28.00	18.00–30.00	25.00	19.00–29.00	25.00	21.00–28.00	0.456
NYHA Class								
	II	36	59.00	38	63.3	42	68.9	0.595
	III	21	34.4	18	30.0	18	29.5	
	IV	4	6.6	4	6.7	1	1.6	
CCI								
	Score (median; IQR)	2	2.0–5.0	2	2.0–4.0	2	1.0–4.0	0.877
EF								
	HFpEF	31	50.8	32	53.3	34	55.7	0.855
	HFmrEF	14	23.0	10	16.7	13	21.3	
	HFrEF	16	26.2	18	30.0	14	23.0	
Self-care maintenance								
	Inadequate (score < 70)	54	88.5	53	88.3	57	93.4	0.564
Self-care management ^¥^								
	Inadequate (score < 70)	47	95.9	51	98.1	34	94.40	0.657
Self-care confidence								
	Inadequate (score < 70)	44	72.1	48	80.00	52	85.2	0.200
Self-care maintenance								
	Score (0–100) (median; IQR)	46.66	29.56–63.33	53.33	35.83–63.33	46.66	36.67–56.66	0.162
Self-care management ^¥^								
	Score (0–100) (median; IQR)	41.00	29.00–55.25	41.28	25.00–57.50	45.11	28.75–55.00	0.566
Self-care confidence								
	Score (0–100) (median; IQR)	50.04	45.56–71.20	52.41	37.82–67.24	51.00	38.92–66.72	0.687

Legend: MI = motivational interviewing; SD = standard deviation; CCI = Charlson comorbidity index; MoCA = Montreal cognitive assessment; NYHA = New York Heart Association; EF = ejection fraction; HFpEF = preserved ejection fraction (left ventricular ejection fraction ≥ 50%); HFmrEF = midrange ejection fraction (left ventricular ejection fraction that ranges from 40 to 49%); HFrEF = reduced ejection fraction (left ventricular ejection fraction < 40%); IQR = interquartile range. ^¥^ In arms 1, 2, and 3, patients with symptoms who filled this part of the SCHIFI were, respectively, 49, 52, and 36 because patients without recent experiences of signs and symptoms were not asked to fill the questions regarding self-care management.

**Table 2 healthcare-11-00773-t002:** Self-care changes and frequencies of patients adequate in self-care during follow-up, and comparisons between experimental arms (arms 1 and 2) and control group.

			Arm 1: MI Only for Patients (n = 61)		Arm 2: MI for Patients and Caregivers (n = 60)		Arm 3: Usual Care for Patients and Caregivers (n = 61)		Arm 1 vs. Arm 3 Cohen’s d (*p*-Value)	Arm 2 vs. Arm 3 Cohen’s d (*p*-Value)
		N	Mean	SD	Mean	SD	Mean	SD		
**Δ in Self-care maintenance scores**										
	T1	182	12.84	11.50	10.81	13.05	2.78	10.33	**0.92 (<0.001)**	**0.68 (<0.001)**
	T2	182	14.60	11.92	10.39	12.01	3.34	11.57	**0.96 (<0.001)**	**0.60 (<0.001)**
	T3	182	17.31	14.71	11.23	13.68	2.73	14.59	**0.99 (<0.001)**	**0.60 (0.001)**
	T4	182	16.37	17.95	7.82	13.89	2.02	11.91	**0.94 (<0.001)**	**0.45 (0.015)**
**Δ in Self-care management scores**										
	T1	114	14.26	19.29	11.50	28.61	10.12	23.06	0.20 (0.788)	0.05(0.586)
	T2	107	11.65	20.41	8.51	20.26	9.4	17.88	0.12 (0.672)	0.05 (0.427)
	T3	108	11.12	18.64	7.79	20.40	9.2	16.72	0.11 (0.663)	0.07 (0.382)
	T4	101	11.50	18.16	12.04	19.10	12.57	23.36	0.05 (0.421)	0.03 (0.462)
**Δ in Self-care confidence scores**										
	T1	182	7.11	14.22	9.04	6.84	3.06	12.84	0.31 (0.101)	**0.58 (0.002)**
	T2	182	7.40	16.06	8.36	9.78	1.79	17.54	0.24 (0.183)	**0.37 (0.043)**
	T3	182	10.08	17.71	9.69	9.07	3.75	16.55	**0.37 (0.044)**	0.32 (0.079)
	T4	182	12.13	13.47	12.56	10.13	3.60	18.55	**0.53 (0.004)**	**0.60 (0.001)**
									**Arm 1 vs. Arm 3** **(*p*-Value)**	**Arm 2 vs. Arm 3** **(*p*-Value)**
**Patients adequate in self-care maintenance (scores ≥ 70)**										
			N	%	N	%	N	%		
	T1		19	31.1	26	43.3	4	6.6	1.000	0.626
	T2		21	34.4	19	31.7	4	6.6	0.134	0.297
	T3		29	47.5	21	35.0	14	23.0	0.313	0.751
	T4		23	37.7	23	38.3	19	31.3	0.507	0.872
**Patients adequate in self-care management (scores ≥ 70)**										
			N	%	N	%	N	%		
	T1		14	30.4	11	21.6	6	14.3	1.000	0.295
	T2		11	24.4	10	20.0	3	8.6	0.342	0.545
	T3		11	25.6	8	17.0	5	12.8	0.454	0.269
	T4		7	15.9	10	22.7	7	21.2	0.342	0.330
**Patients adequate in self-care confidence (scores ≥ 70)**										
			N	%	N	%	N	%		
	T1		19	31.1	17	28.3	10	16.4	0.261	**0.003**
	T2		18	29.5	19	31.7	9	14.8	**0.015**	**0.023**
	T3		20	32.8	21	35.0	12	19.7	**0.046**	**0.011**
	T4		27	44.3	23	38.3	14	23.0	**<0.001**	**<0.001**

Legend: SD = standard deviation; *p*-values in bold are <than α (5%).

## Data Availability

Data are available from the corresponding author upon reasonable request.

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
