# Peer review of "A Single-Center, Randomized Controlled Trial to Test the Efficacy of Nurse-Led Motivational Interviewing for Enhancing Self-Care in Adults with Heart Failure"

_healthcare, 2023, doi:10.3390/healthcare11050773_

Round 1
Reviewer 1 Report
The aim of the study is to inquire about the role of nurse-led motivational interview in the clinical management of adults with heart failure in what concerns improving self-care maintenance, management and confidence.
The study covers a gap in medical literature since nurse-led interventions are less focused and studied. The management of heart failure patients implies education of patients and caregivers that translate into clinical beneficial effects. Self-care and its determinants are less studied in heart failure patients.
1. However, the same group published several articles on the subject including a similar study:
Motivational interviewing to improve self‐care in heart failure patients (MOTIVATE‐HF): a randomized controlled trial
ESC Heart Failure
2020-06-28 | Journal article
DOI:10.1002/ehf2.12733
Contributors: Ercole Vellone; Paola Rebora; Davide Ausili; Valentina Zeffiro; Gianluca Pucciarelli; Gabriele Caggianelli; Stefano Masci; Rosaria Alvaro; Barbara Riegel
There is a significant overlap with the aforementioned published study.
2. Moreover, some of the authors of this manuscript published a meta analysis on the subject
Ghizzardi G, Arrigoni C, Dellafiore F, Vellone E, Caruso R. Efficacy of motivational interviewing on enhancing self-care behaviors among patients with chronic heart failure: a systematic review and meta-analysis of randomized controlled trials. Heart Fail Rev. 2022 Jul;27(4):1029-1041. doi: 10.1007/s10741-021-10110-z. Epub 2021 Apr 17. PMID: 33866487.
The present study does not bring new data.
3. The methodology is sound and well documented. However, since the inclusion ended May 2020 after the start of Covid 19 Pandemic, one should ask how the direct health intervention provided by the nurse impacted in this regard.
4. Another issue is the overlap between the inclusion period of time between the two studies: June 2014 to October 2018 in MOTIVATE-HF Study and May 2017 to May 2020 for the present study. Since the methodology is very similar, is it an overlap or a multicenter study?
5. Inadequate self-citation. E.g. 6 self-citation of the first author Dellafiore, F. and 11 self-citation of Vellone, E.
Reference 19 is not valid.
6. Taking all the aforementioned arguments into consideration, I would recommend the authors to rewrite the discussion section being more specific about the novelty of the study and the results compared to previous studies.
Author Response
Please find our point-by-point responses in the attached file.

Reviewer 2 Report
The manuscript reports about a single centre randomized trial about the effect of motivational interviewing in self-care of patients with chronic heart failure.
Both the design of the study and the text are very accurate, compliant with the CONSORT statement (that the authors could quote to enforce the validity of their results), and I congratulate with the authors for this study.
I have only two observations, referred to the interpretation of data
1. page 14, lines 309-10: "the nurse-led MI performed with scheduled recurrences over time produces stable effects in improving self-care maintenance over time". Since there is not a group who had only a single, not repeated intervention, you cannot imply that the stability is DUE to the repetition. It is reasonable and "likely" to be, but it must be proved
2. page 14, lines 313-14 and page 15, line 326: "The more stable effects on improving self-care maintenance showed in Arm 1 over Arm 2" Maybe I missed the data, but there is no comparison between Arm 1 and Arm2. If so (and if it is not so, and I missed the data, these data must be put forward in a clearer way, for the benefit of the reader), you should at least change the statement in the form of an hypothesis for further research. The interesting discussion in lines 327 and following could be the rationale.
A minor issue is at page 2, line 63, where you use the abbreviation MI, which has been disambiguated only in the Abstract, please add the full name
Author Response

(The authors gave the same response as above.)
